# Effect of Geometric Arrangement on Mechanical Properties of 2D Woven Auxetic Fabrics

**Arif Ali Shah** [1,*]**, Muhmmad Shahid** [1,*] **, Naveed Ahmad Siddiqui** [2]**, Yasir Nawab** [3] **and Mazhar Iqbal** [2]

1    School of Chemical and Materials Engineering, National University of Sciences & Technology, Islamabad 44000, Pakistan
2    Centres of Excellence in Science & Applied Technologies (CESAT), Islamabad 44000, Pakistan
3    Dean School of Engineering and Technology (SET), National Textile University, Faisalabad 44000, Pakistan
*    Correspondence: arifphd.scme@student.nust.edu.pk (A.A.S.); mshahid@scme.nust.edu.pk (M.S.)

**Abstract:** Textiles-fibres, yarns and fabrics are omnipresent in our daily lives, with unique mechanical properties that fit the design specifications for the tasks for which they are designed. The development of yarns and fabrics with negative Poisson's ratio (NPR) is an area of current research interest due to their potential for use in high performance textiles (e.g., military, sports, etc.). The unique braiding technology of interlacement for preparation of braided helically wrapped yarns with NPR effect with later development of auxetic woven fabric made it possible to avoid the slippage of the wrapped component from the core. The applied geometrical configuration and NPR behaviour of the braided helical yarn structure with seven different angles comprising of monofilament elastomeric polyurethane (PU) core with two wrap materials that include multifilament ultra-high molecular weight polyethylene (UHMWPE) and polyethylene terephthalate (PET) fibres were investigated and analysed. The mechanically stable 2D woven textile auxetic fabrics (AF) with various weave patterns such as 2/2 matt and 3/1 twill were developed from the auxetic yarn with PU elastomer core having maximum NPR effect of −1.70 using lower wrapped angle of 9° to study and compare their mechanical responses. The auxetic yarn was used in weft direction and multifilament UHMWPE yarn in warp direction, using semi-automatic loom. Auxeticity of AF was analysed and its various mechanical properties such tensile strength, impact energy absorption, in-plane, and out-of-plane auxeticity, and puncture resistance were studied. Higher energy absorption of 84 Nm for matt fabric was seen compared to twill fabric having an energy of 65 Nm. The puncture resistance capability of matt fabric was better than twill fabric. While twill fabric exhibited better auxetic effect in both in-plane and out-of-plane mode compared to matt fabric. In short, both the twill and matt design AF's showed unique characteristics which are beneficial in making various protective textiles such as protective helmets, bullet proof shields, cut resistance gloves, blast resistant curtains, and puncture tolerant elastomeric composites.

**Keywords:** negative Poisson's ratio; yarn; woven auxetic fabric; twill; matt; puncture resistance; protective textile

## 1. Introduction

Auxetic materials are compose a new class of materials that behave contrary to the normal materials, which contract in transversal direction when expanded longitudinally [1]. These materials are better described by the Poisson's ratio having negative values in contrast to normal materials [2]. Auxetic materials are categorized both as natural and synthetic. Auxetic honeycomb (AHC) exhibit best auxetic nature amongst all auxetic geometries [3]. Natural Auxetic materials are human body parts such as bone tissues, tendons, skin [4], cubic crystals, cow skin, and cat teats [5]. Foams, fibres, yarns, fabrics, and composites are classified as synthetic auxetic materials. Auxetic foams have higher load bearing capacities and formability than conventional foams [6]. Therefore, auxetic foams are categorized as

good scaffold candidates for tissue engineering and spinal implants [7]. Auxetic materials have also been proved to have better properties compared to conventional material such as good sound absorption, better elongation and shear resistances, high flexural and impact strengths [8]. Auxetic materials have much higher energy absorption factor, e.g., vibration damping as compared to conventional materials.

Auxetic textiles have attracted great attention in the recent era. Researchers have tried to induce the auxetic behaviour from fibre to fabric level with auxetic geometries in the textile structures [9–14]. Auxetic fibres for these textiles were developed through the modification of the conventional melt spinning route. Auxeticity is imparted in nylon, polypropylene, polyester, and UHMWPE, etc., at fibre spinning level [15–18]. Auxetic fibres-based composites showed superior mechanical properties because NPR induces self-sticking effect and higher pull-out resistance in auxetic fibres [15]. However, the excessive cost of auxetic fibres is a major limitation in their commercialization.

Auxeticity was therefore introduced at yarn level by simple wrap spinning technique. The high modulus yarn was wrapped over low modulus yarn, to make helical auxetic yarns (HAY). The low modulus of HAY limits their high-speed processing on conventional weaving machines. Woven fabric and corresponding auxetic composites made from auxetic yarns also showed NPR and superior mechanical properties [19–22]. Double helical auxetic yarn (DHY) was used to produce plain woven fabric as reinforcement for composite materials [19]. The meta-aramid fibre was used as warp while the DHY was used as weft. The yarns in the woven fabric overlapped each other in out-of-plane manner. Due to this overlapping, the fabrics showed auxetic behaviour in thickness direction only.

Woven fabrics of two types produced from HAYs with wrap angle 45° in the warp direction exhibited an in-plane NPR in a strain range of 15–40% and a positive Poisson ratio for all values of applied strain [23]. It was seen that higher values of elasticity or crimp in the yarn produced the in-plane NPR of the fabric. Plain, 2/2 twill and 8-end satin fabrics were woven from HAY in weft direction with the maximum NPR effect exhibited by plain weave [12]. Auxeticity when induced at the fabric level had several benefits such as continuous manufacturing, low cost, and no material limitation.

Knit structures being flexible and versatile are preferred as Auxetic structures, but these structures having problems of low structural stability, are combined with weaving to make the stable co-weave auxetic composite structures. Such structures are developed on special machines because conventional machines cannot develop such structures. This technology is used to produce 3-D auxetic knit auxetic structures with increased stability, but limited production [24], because these structures could only be produced on specialized machines. Numerical and mathematical modelling of various geometrical parameters, and their composites were developed and tested for fracture toughness, resistance to indentation, energy absorption and impact applications [25–29]. The auxetic composites are used in structural and high-tech applications due to their higher shear strain resistance properties, e.g., bridges, buildings, and airplanes. Auxetic composite materials can be made either from conventional components via specifically designed internal structural configurations or from auxetic reinforcements. The failure of conventional composites during mechanical loading can be improved significantly by replacing them with auxetic composite specifically for high impact applications. The auxetic effect enhances the flexural and impact properties with no effect on the tensile properties.

The auxetic fabrics (AF's) play significant role in the field of medical and body wearing applications due to their compatibility with body profile, such as during knee/elbow bending or other medical applications: there is a longitudinal stretch on fabric, while at the same time, thigh muscles are swelling. AFs have an anisotropic nature, lower stiffness, and higher formability. The higher sensitivity and pore opening effect of AFs makes them highly demanding in medical textiles for example the bandage used for swelling wound care. Non-auxetic bandages develop a high pressure on the wound and decrease in porosity thus the wound heals slowly. This is also a critical problem in compression garments when used for burning care after plastic surgery. Conventional fabrics are much less sensitive

than auxetic fabric when used for smart biomedical devices [30]. AFs with NPR effects, exhibit a higher change in the surface area, and therefore have higher changes in resistance or sensitivity.

The yarns for wearable auxetic Auxetics are produced from simple wrap-spinning techniques. Wrap spun auxetic yarns are developed by wrapping a highly stiff fine yarn on the coarse low-modulus core yarn. However, as auxetic yarns are based on elastic materials, their preliminary extension is extremely high which creates hurdle while processing them on high-speed machines. Therefore, the fabric structure level auxeticity offers low cost and continuous manufacturing process in addition to minimizing the use of synthetic materials. Weft and warp-knitted structures are utilized for the development of AFs due to their flexibility, versatility in design, and high-speed manufacturing processing. Knitted AFs have low structure stability but have superior comfort properties with respect to pressure distribution and body shape adoption [31,32]. Knit structures with NPR effect made of conventional elastic yarns of low stiffness thick filament core and a high stiffness wrap were also developed [9].

Two-dimensional woven AFs developed with our proposed materials have not yet been reported. Such AFs can fulfil the need of protection wearing textile, as these are developed by a simple technique and are expected to be more stable with NPR effect. Therefore, this research is focused to study the development and response of structural changes of auxetic yarn and geometric patterns of fabric structures with largest NPR value and then evaluate them for optimized structural parameters. Uniaxial tensile and low velocity impact tests are conducted to evaluate and compare the yarn and fabric mechanical properties. These properties were compared to ensure suitability for protective textile applications, while auxeticity was induced by orienting yarns in distinctive designs. The difference of material in warp and weft dimensions makes the auxeticity different along principal directions.

## 2. Materials and Experimental Method

### 2.1. Auxeticelopment of Auxetic Yarn

The materials and methodology utilized for development of auxetic yarn is same as we reported in our previous work [33]. Various manufacturing parameters for producing auxetic yarn are shown in Table 1. Seven yarn samples, S1 to S7 composed of poly urethane (PU) elastomer as core and a combination of UHMWPE and PET as wrap, were made using seven different twisting speeds in the range of 70 to 154 rpm on the main motor with a fixed delivery speed of 1 rpm at the take-up spool motor. The different braid angles of the two wraps with core yarn measured with Image J software are shown in Figure 1.

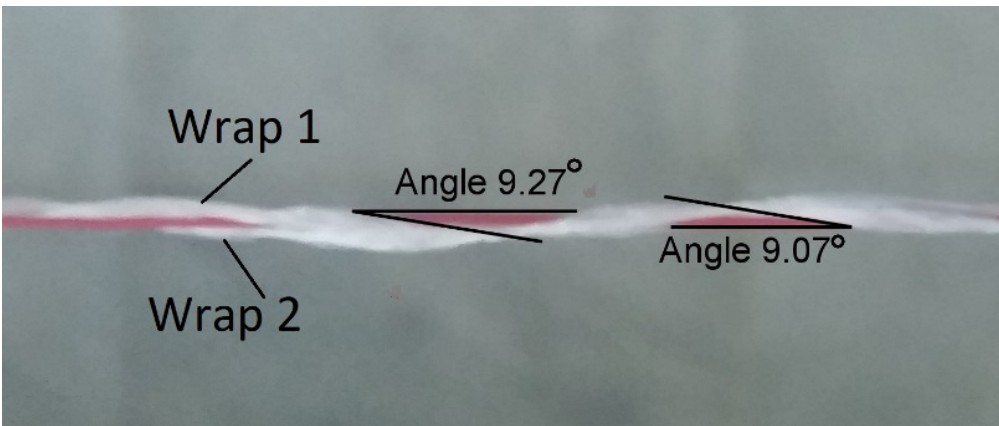

**Figure 1.** Measured angles of twists for both the wraps with PU core.

**Table 1.** Manufacturing parameters of auxetic yarn [33].

|  | Twisting Speed (rpm) | Delivery Speed (rpm) | Braid Angle (°) | Poisson's Value |
|---|---|---|---|---|
| S1 | 70 | 1 | 9 | −1.70 |
| S2 | 85 | 1 | 11 | −0.86 |
| S3 | 100 | 1 | 13 | −0.11 |
| S4 | 115 | 1 | 17 | −0.08 |
| S5 | 130 | 1 | 19 | −0.03 |
| S6 | 145 | 1 | 21 | −0.04 |
| S7 | 154 | 1 | 23 | −0.01 |

### 2.2. Development of Woven Fabric

The auxetic yarn S1 was used in the weft direction while UHMWPE multifilament yarn was used in the warp for fabric development from the auxetic yarn. Samples with 24 ends per inch and 24 picks per inch were woven on a semi-auto dobby weaving sampling loom (SGA 598, China). Photographs of the dobby weaving machine and fabric sample during fabrication stage are shown in Figure 2a,b.

Characterization of the Woven Fabric

Single layer of fabric was used for characterization to avoid irregularity in the fabric surface texture. Various warp wise and weft wise tensile tests, in-plane, and out-of-plane auxeticity tests, puncture resistance, and impact energy absorption tests were performed to compare the properties in two dissimilar weave design fabric patterns. A square unit cell of (1 inch × 1 inch) marked on the fabric strip was used for measurement of lateral and longitudinal strains of in-plane auxeticity of AF. Pictures were captured with high resolution camera at a fixed interval of 5 s with a fix strain. The changes in the dimensions of the square along the axial and transverse directions were analysed by Image J software.

The tensile tests were performed with LLOYD LRX Plus Ametek, (USA) tensile machine shown in Figure 3a, the sample sizes were kept 100 mm × 60 mm, and speed during the test was 100 mm/min. Three samples each for matt and twill design in the warp and weft direction were evaluated. Fabric tensile properties such as breaking and elongation along the warp and weft direction were also noted. Out-of-plane auxeticity was determined by clamping the fabric samples in a tensile testing machine equipped with a dial indicator for measuring the thickness of the fabric at different points as shown in the Figure 3b. The thickness of the fabric was measured at the start of each experiment in all samples. These fabric samples were clamped in the fixtures of tensile machine, loaded gradually and variation in thickness of the fabric with respect to the longitudinal change in length for a fix distance of 20 mm was recorded. The experiments were repeated two times for both the matt and twill design fabric samples in warp and weft directions, respectively.

The puncture resistance test of the samples was performed as per standard BS EN-388. Four samples each were assessed for matt and twill pattern fabrics. The same LLOYD Instruments LRX Plus, tensile testing machine with NexGen software was used for puncture resistance tests. The gauge length was kept 100 mm and the test speed was 100 mm/min. Finally, the fabric energy absorption setup is shown in Figure 3c. The gun was fixed at distance of 6 ft. from the fabric holding setup. The fabric sample size was kept 254 mm × 305 mm. The difference of the initial energy of fired air gun shot, and the final energy absorbed by the fabric produced the energy absorbed by the fabric. The kinetic energy (K.E.) of the gun shot before and after passing through both types of fabrics was calculated from the basic energy equation given below.

$$\text{K.E.} = \frac{1}{2}mv^2 \tag{1}$$

where m is the mass of the gun shot, and $v$ is the velocity of fire.

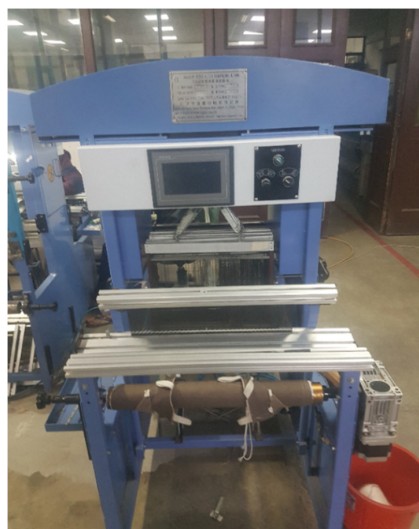

(**a**)

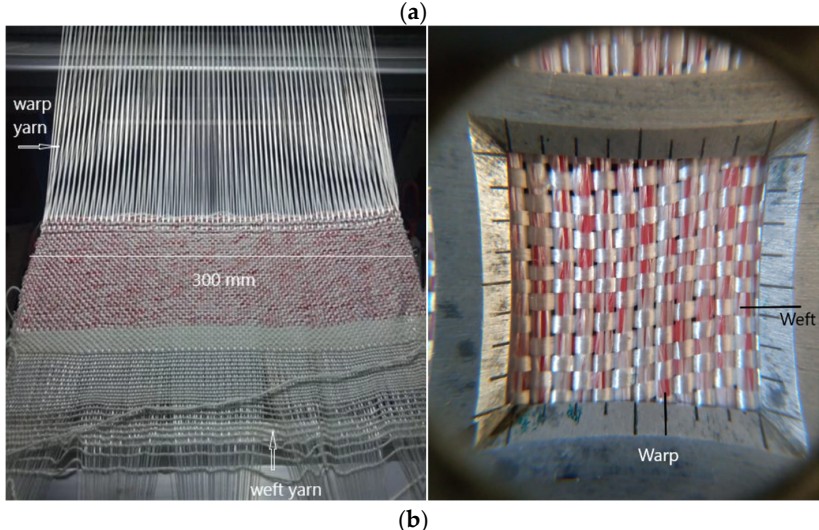

(**b**)

**Figure 2.** (**a**) Semi-automatic dobby weaving Loom used for making fabric samples in this study. (**b**) Fabric sample during production stage. Closer view of the fabric is shown on the right clearly showing warp and weft yarns.

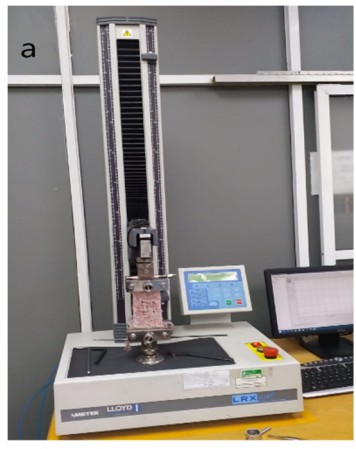
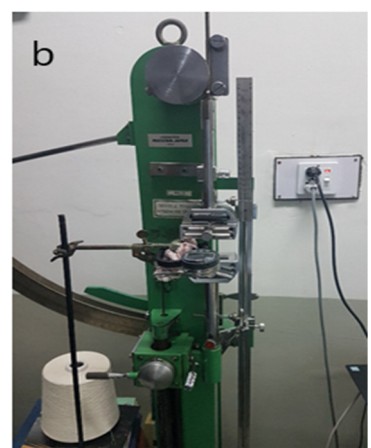
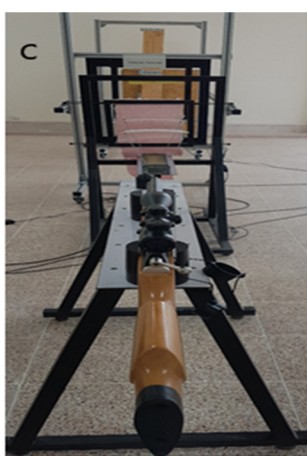

**Figure 3.** Fabric tensile testing machines: (**a**) Tensile and puncture resistance machine; (**b**) Out-of-plane auxeticity test machine; (**c**) the energy absorption setup.

### 3. Results and Discussion

#### 3.1. The NPR Effect

Figure 4a depicts the graph of largest NPR values corresponding to the respective braid angles for samples S1 to S7 as obtained from Table 1. Comparison amongst the seven yarns revealed that yarn S1 with maximum NPR value of −1.70 had a better auxetic behaviour than S7 with NPR of 0.01. As we focused on the NPR value of the yarn to develop the fabric, we chose the yarn S1 for development of the auxetic fabric. The standard deviations for all these samples are shown in Figure 4b.

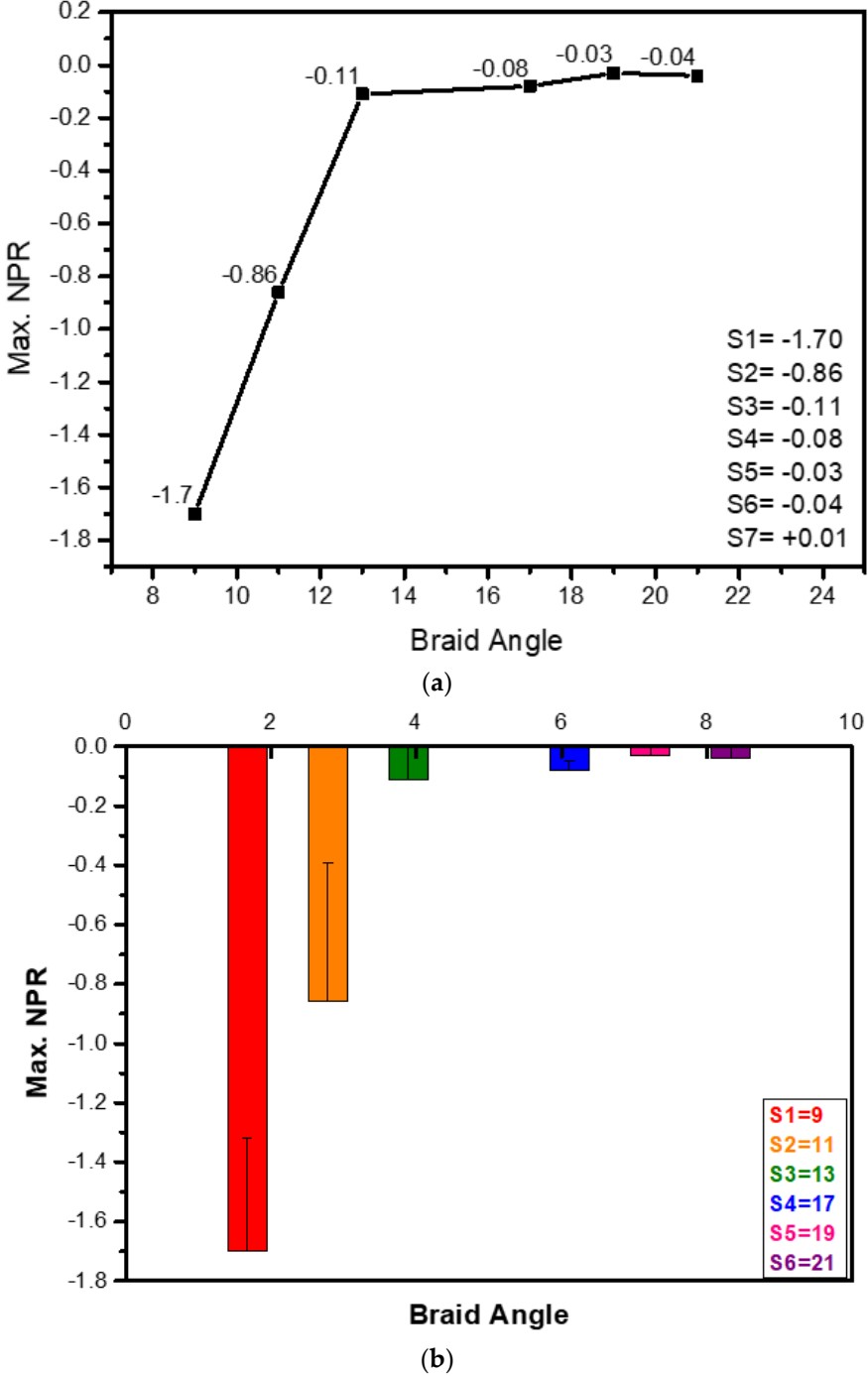

**Figure 4.** (**a**) Maximum NPR vs. braid angle for samples S1 to S7. (**b**) Error bars for maximum NPR values versus braid angle.

### 3.2. Tensile Tests in Warp and Weft Direction

The tensile test results of woven Afs exhibit high stability and strength with excellent properties and deformation behaviour. All the matt and twill pattern fabric samples are identified by M1-M4, and T1-T4, respectively, along the weft and warp directions in our experiments as shown in Table 2. A larger variation in load-extension curves of matt weft fabric was observed as compared to warp direction. The samples of matt fabric exhibited 15% more force with 60% more elongation along the warp direction compared to the samples in weft direction as shown in Figure 5a. This variation is due to the weaving of the fabric, i.e., during weaving stage, the tension, and the density of the fabric in warp direction were controlled automatically by machine, while the parameters in weft direction were controlled manually, which affected the weave quality and in turn the fabric response. The material elasticity and the applied tension during warp and weft weaving, caused the fabric density to increase when the fabric was taken off the loom.

**Table 2.** Sample identification for woven fabric with different geometrical design.

| Sr. No. | Sample ID | Weave Design | Weave Direction | Braid Angle of Yarn |
|---|---|---|---|---|
| 1 | M1 | | Weft | 9° |
| 2 | M2 | 2/2 Matt | | 9° |
| 3 | M3 | | Warp | 9° |
| 4 | M4 | | | 9° |
| 5 | T1 | | Weft | 9° |
| 6 | T2 | 3/1 Twill | | 9° |
| 7 | T3 | | Warp | 9° |
| 8 | T4 | | | 9° |

Comparing the variations between the warp and weft directions for the twill fabric samples shown in Figure 5b, the weft twill samples T1-T2 exhibited twice the loads (about 4.6–5.0 KN) than that of warp twill samples T3-T4 which bear 2.5–3.0 KN. This is because the warp yarns were repeatedly loaded during weaving by beat-up, while the weft yarns were manually weaved and hardly loaded. The pretension of the warp yarns also has influence on the results. On the other hand, the twill warp fabric exhibited twice the extension compared to the weft fabric due the presence of auxetic yarn in the weft direction which restricted the movement of the yarn in weft direction.

The crimps in the yarns of the fabric also play a particularly key role. All the weave designs have their own unique crimp percentages depending on the interlacement pattern and float length. The crimp is maximum for plain-woven fabric, gradually decreases for twill and is least for the matt woven fabric. The crimped yarns become straighten when subjected to tensile loads. A further increase in the load after straightening of yarns cause them to break. Initially lower force is needed to remove the crimp in fabric structure, and afterwards they break at higher loads. Therefore, having least crimp in structure, matt woven fabrics withstand a maximum load with a maximum extension of yarns before breakage in warp direction. These results are in accordance with the reported literature [10].

The twill and matt design fabrics when compared to each other in both the principal directions exhibited the behaviour as shown in Figure 6a,b. The twill fabric exhibited a lower load with low modulus and a 50% more extension in warp direction as compared to the matt fabric. Higher initial elongation renders lower initial modulus, but when the yarns became straight, the modulus increased at a faster rate. There was an outward lateral pull on the weft when the fabric sample was stretched longitudinally and vice versa due to the presence of auxetic yarn in this direction. The twill weft direction observes more force due to the 3/1 twill geometry which observed less restriction offered by the auxetic yarn in this direction. This also revealed that if stress was applied on the warp, some energy would be absorbed in weft stretching, hence weft contributed more to warp strength and energy absorption. Twill fabric has crimps higher than the matt fabrics; lower initial loads are required to remove their crimps. Therefore, matt weave samples observed maximum

load at lower extensions in warp direction. In the weft direction, the crimps in the matt design increase due to the auxetic yarn. The lower loads and more elongation at break make the matt fabric leading to superior protective nature of woven auxetic fabrics with more energy absorption capabilities.

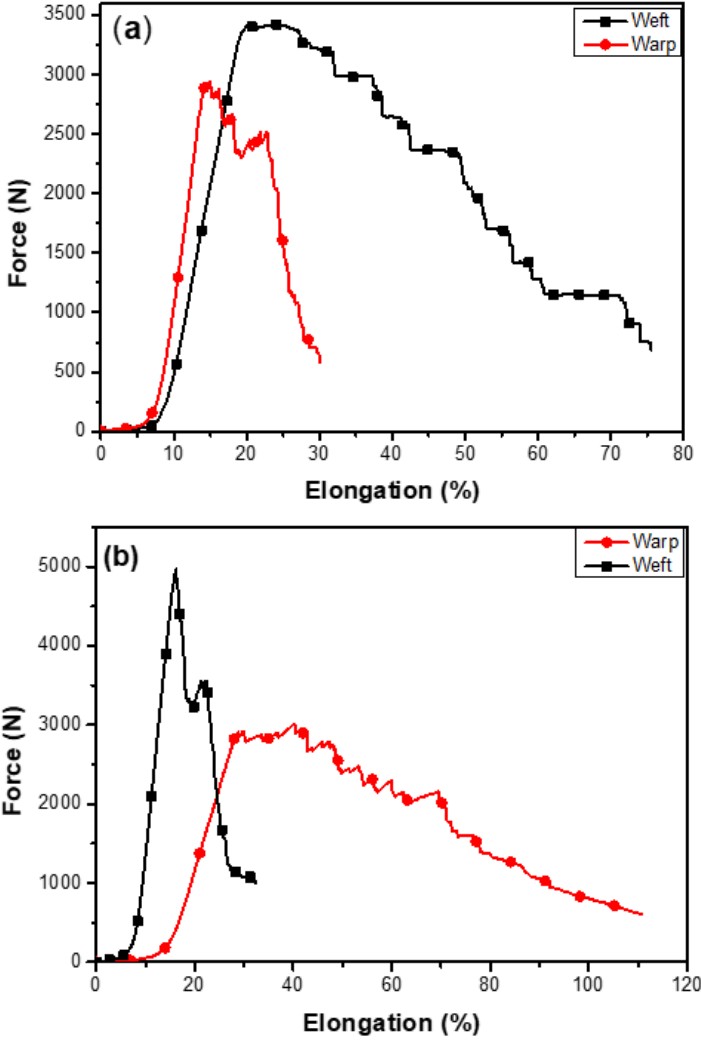

**Figure 5.** Comparison of tensile properties of auxetic woven fabric. (**a**) Matt weft and warp, and (**b**) twill weft and warp.

### 3.3. In-Plane Auxeticity

The In-plane auxeticity of woven fabric in respective directions was measured with the help of software Image-J by incorporating changes in lateral and longitudinal strains of a small square cell marked on the fabric strip using the tensile testing machine, shown in Figure 7a,b. The in plane-auxetic behaviour of the woven fabric was smaller compared to the auxetic yarns due to the crimp interchange of the fabric. For example, upon loading the woven fabric in the warp direction, the warp yarn crimp reduced while the crimp in the weft yarn was forced to increase, resulting in narrowing down of the fabric in the weft direction. At higher fabric density, the auxetic yarns pushed out, but with minimal fabric expansion because of yarn deformation in the transverse direction and the crimp interchange as auxetic yarn would not influence the fabric dimension because the increase in the outer contour diameter of the yarn was still smaller than the gap between two adjacent yarns.

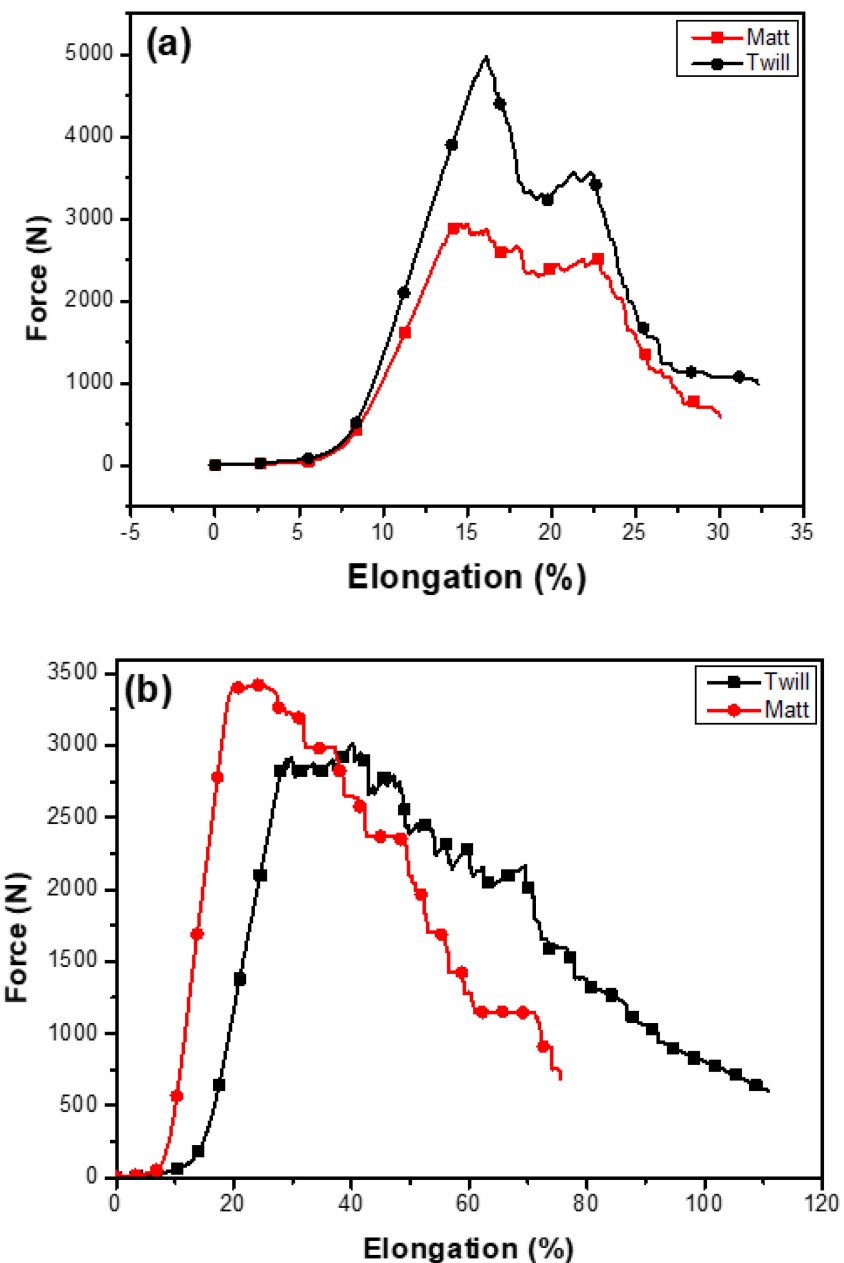

**Figure 6.** Comparison of tensile properties of matt and twill fabrics (**a**) weft direction and (**b**) warp direction.

In AFs, the mutual support of warp and weft yarn makes them shrink and expand alternately, thus in relaxed form of the fabric, the differential shrinkage enables the unit cell of the fabric structure to contract non-uniformly, and the fabric undergoes various levels of shrinkage at different sections resulting in folds formation. When stretched, the unfolding of the folded area starts not only in the stretch direction, but also in the transverse direction, giving rise to the NPR effect. Therefore, the NPR effect results from the interplay between the interlacement pattern of warp and weft in addition to the fabric geometry such as plain, matt or twill, the different stretch properties of the elastic and nonelastic yarns, i.e., the core and the wrap materials, and finally the deformation mechanism of the fabric [10]. Under tensile loading, the axial part of the unit cell tends to expand the lateral portion, resulting in the lateral expansion of initial small-width portion, and increase in its width as well, thus an overall lateral expansion of the fabric occurs. During this process, the warp yarns supply the required support for the opening and reorienting of the yarns in the weft direction.

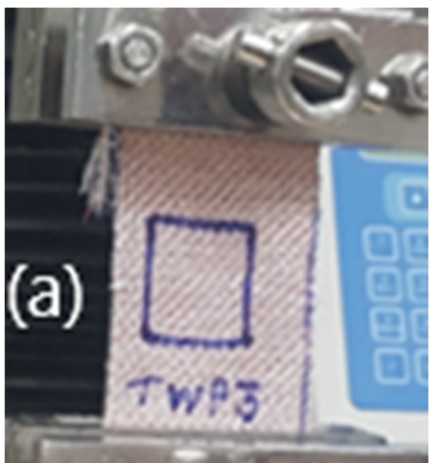 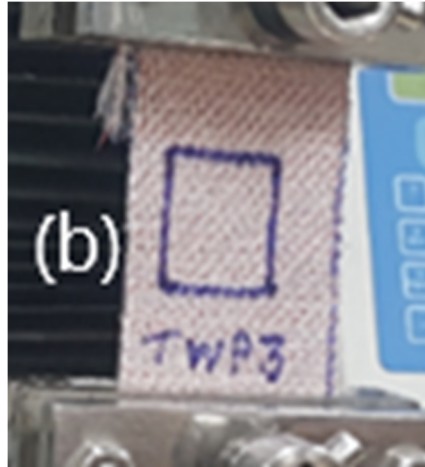

**Figure 7.** Changes in the dimensions of the unit square cell (**a**) initial relaxed position, (**b**) stretched position.

### 3.3.1. Warp Wise NPR Characterization

The highly interlaced structure in warp direction supports the high float structure which is pushed laterally outward upon application of longitudinal loads. This outward expansion causes an increase in dimensions of high float structure, thereby causing an outward expansion in whole fabric structure, thus observing auxetic effect in the fabric. The initial and final difference in dimensions of the square cell in relaxed and stretched condition analysed with Image J software and using equation 2 give rise to the auxetic effect. Three experiments each were performed warp wise for matt and twill design fabrics. The force was applied up to a fix elongation of 10 mm for all the samples. The initial dimensions of the square cell are 25.4 mm × 25.4 mm in relaxed condition. Changes were observed in both the matt and twill fabrics along the principal directions of the fabric upon application of force in the warp direction as shown in Figure 8a–h. The initial distance between points of unit cell, the changes in lengths and corresponding warp-wise NPR values are shown in Table 3.

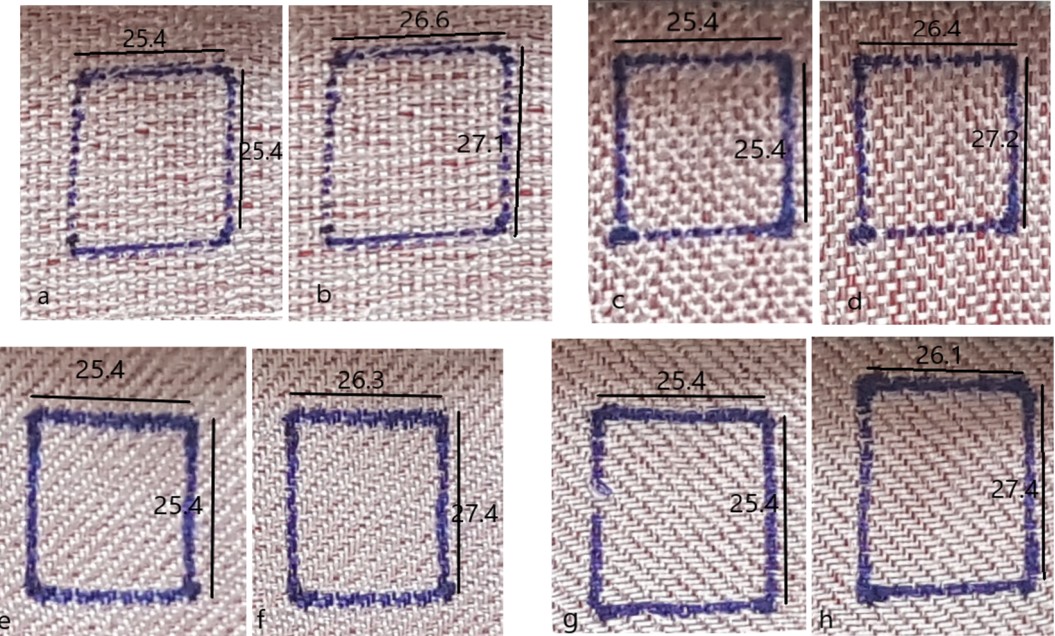

**Figure 8.** Changes in longitudinal and lateral directions for all fabrics samples. (**a**,**b**) Matt warp, (**c**,**d**) matt weft, (**e**,**f**) twill weft, and (**g**,**h**) twill weft.

**Table 3.** NPR values associated with corresponding strains.

|  | Axial ΔL (mm) | Lateral ΔL (mm) | Axial Strain, $\varepsilon_a$ | Lateral Strain $\varepsilon_l$ | Poissons Ratio $\upsilon = -\varepsilon_l/\varepsilon_a$ |
|---|---|---|---|---|---|
| Matt warp | 5.70 | 2.41 | 0.04 | 0.03 | −0.67 |
| Twill warp | 7.11 | 2.97 | 0.04 | 0.04 | −0.83 |
| Matt weft | 2.85 | 1.93 | 0.05 | 0.03 | −0.69 |
| Twill weft | 3.10 | 2.22 | 0.03 | 0.03 | −1.00 |

### 3.3.2. Weft Wise NPR Characterization

The small floats of warp and weft as in matt design fabric result in firm woven yarns with lesser mobility compared to twill fabrics. Consequently, the matt fabric undergoes lesser shrinkage as compared to the twill fabrics thus creating differential shrinkage within the fabric structure. In the relax condition of the fabric, this differential shrinking causes a zig zag pattern of yarns in the warp direction. Upon stretching along weft direction, the weft shrinkage is minimized, and the bending in warp yarn is removed to become straighten. Resultantly, weft wise NPR is observed due to the warp straightening phenomenon with an increase in fabric dimensions along the warp and weft. The weft wise NPR values associated with corresponding strain values are also tabulated in Table 3.

The NPR in warp direction is relatively high with lower axial strain values, as compared to the weft direction, respectively, for both the fabrics. The fully relaxed warp fabric can be stretched easily at the initial stage. However, as lateral expansion increases, it requires more strain to further open the structure due to the resistance offered by the auxetic yarn thus decreasing Poisson's ratio at higher axial strains.

The matt fabric faces the resistance in mobility caused by the frictional forces at the yarn crossover points of the auxetic yarn in the weft direction at higher strains 0.05 thus exhibits at lower NPR effect −0.69. On the other hand, the twill fabric having high floats, and lower cross over points, faces lower resistance offered by the auxetic yarn present in the weft direction, can be stretched easily thus having higher NPR effect −1.0 even at comparatively lowest strain values 0.03. The NPR effect is plotted against the fixed displacement of 10 mm for both the fabrics and is shown in Figure 9. The corresponding error bars graph for in-plane NPR effect is shown in Figure 10.

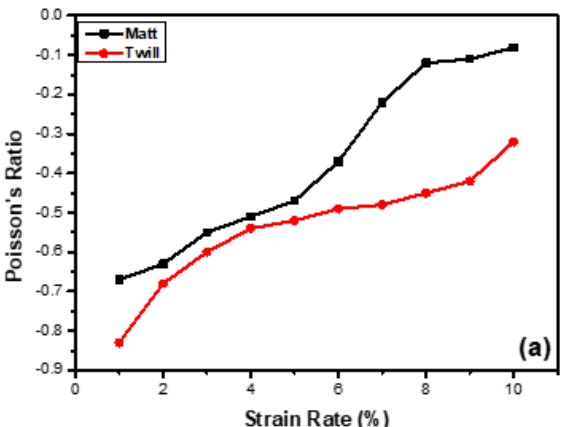
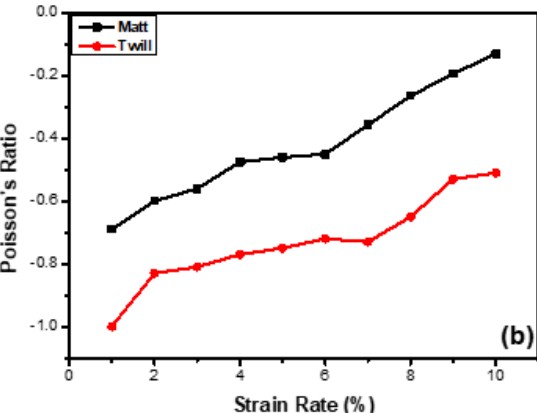

**Figure 9.** Comparison of in-plane NPR for both fabrics, (**a**) warp-wise, (**b**) weft-wise.

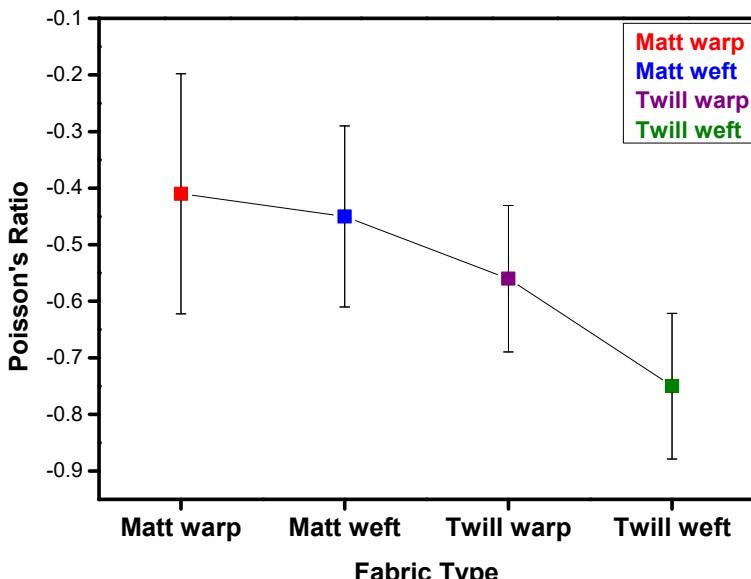

**Figure 10.** In-plane Auxetic effect for all fabric samples having Poisson's ratio with error bars.

The shifting of NPR from higher to lower values due to tensile deformation mechanism involves three stages. Initially, upon stretching in any direction, the yarn shrinkage at loose weave sections rearranges, and the tensile yarns tend to become straightened, followed by the opening of the folded/wrinkled areas in the transverse direction immediately at lower tensile strains, thus resulting in higher initial NPR effect. In the second phase, the yarns move apart from each other, and the tensile strain increases with a decrease of NPR effect. The third phase correspond to a further outward shift where the tensile yarns move towards straight form, a more ordered and consolidated orientation yarn system is achieved with the frictional forces at the yarn cross over points, which restricts further increase in transverse direction and the tensile force is consumed by the tensile strain with the decrease in the NPR effect. This behaviour is continued until the frictional forces at the crossover points are overcome, and the yarns slip over each other after which the fabric behaves conventionally with positive Poisson's effect. It is obvious from the results that the twill woven fabrics exhibit the maximum value of NPR. This is contrary to the stability of the structure, i.e., based on crimp percentages, the structure with least crimp such as matt fabric must have a maximum NPR effect upon subjecting to tensile loads. However, twill fabric exhibited maximum NPR with higher crimp percentage thus has less stable structure at fixed elongation. Further work is suggested on this unusual behaviour of the fabrics.

The auxetic effect in a woven structure also depends on the float length of warp and weft yarn in the structure. If yarn float length is more in a woven fabric, yarns will be loose because of less interlacement points in a repeating unit which in turn, make a structure less compact such as twill fabric. Due to the loosely woven structure, the yarns cannot apply pressure properly on neighbouring yarns. Conversely, if the yarns are tightly packed such as in plain or matt woven fabric, their movement will be restricted and no behavioural change under load will be observed [34]. Therefore, the auxetic yarn in warp direction of these fabric needed optimum compactness, to facilitate the interchanging phenomenon of wrap and core.

### 3.4. Out-of-Plane Auxeticity

Through the thickness auxeticity teste were performed to predict the out-of-plane auxeticity for both the matt and twill design fabrics. In-plane NPR (increase in width) is valid for samples having smaller widths. Wider width flexible textile structure fabrics could not be evaluated for in-plane auxeticity, due to accumulation of wrinkles at certain points thus increasing the fabric thickness. In contrast with other out-of-plane auxetic materials such as foams [35]

and 3-D fabrics [36], this increase in thickness is due to the folding of non-accommodated width-wise expansion. An increase in fabric thickness is observed in all samples due to foldability of the flexible fabric in the form of wrinkles, which cause an improved NPR effect. To predict the increase in thickness or the out-of-plane NPR effect, a digital fabric thickness tester was used. The lengths and thicknesses of the matt and twill woven fabric in warp and weft directions against various strain percentages are given in Table 4. The initial lengths for both fabrics were 5 inches in warp and 3 inches in weft direction. The initial thickness of the matt samples in relaxed condition along both principal directions was measured as 1.33 mm. Similarly, the initial thickness along both direction for twill samples was 1.02 mm. Figure 11a indicates the relationship between Poisson's ratio and the strain percent along warp direction for both matt and twill design through the thickness woven fabric. The thickness increases with the increase in length due to the reinforcing behaviour of binding yarns which try to become straighten by pushing the yarns outwards apart thus resulting in the increased thickness of the fabric. After a threshold strain rate, the NPR value decreases with decrease in the thickness upon further application of load, because the yarns become straightened and no further auxeticity is observed.

**Table 4.** Out-of-plane thicknesses with NPR and corresponding strain rates.

| Sample | Initial Length (mm) | Final Length (mm) | Initial Thickness (mm) | Final Thickness (mm) | Strain % | Poisson's Ratio |
|---|---|---|---|---|---|---|
| Twill warp | 127 | 130.5 | | 1.10 | 2.7 | −2.89 |
| | | 135.8 | | 1.35 | 6.9 | −4.68 |
| | | 140.3 | | 1.59 | 10.4 | −5.38 |
| | | 145.8 | 1.02 | 1.85 | 14.8 | −5.49 |
| Twill weft | 76.2 | 78.4 | | 1.12 | 2.9 | −3.38 |
| | | 80.9 | | 1.33 | 6.2 | −4.90 |
| | | 84.3 | | 1.62 | 10.6 | −5.54 |
| | | 87.2 | | 1.84 | 14.4 | −5.60 |
| Matt warp | 127 | 129.2 | | 1.38 | 1.7 | −2.17 |
| | | 132.7 | | 1.44 | 4.5 | −1.82 |
| | | 136.1 | | 1.50 | 7.2 | −1.77 |
| | | 139.8 | 1.33 | 1.55 | 10.1 | −1.64 |
| Matt weft | 76.2 | 77.4 | | 1.39 | 1.6 | −2.80 |
| | | 79.7 | | 1.49 | 4.6 | −2.60 |
| | | 82.1 | | 1.58 | 7.7 | −2.44 |
| | | 84.3 | | 1.68 | 10.6 | −2.50 |

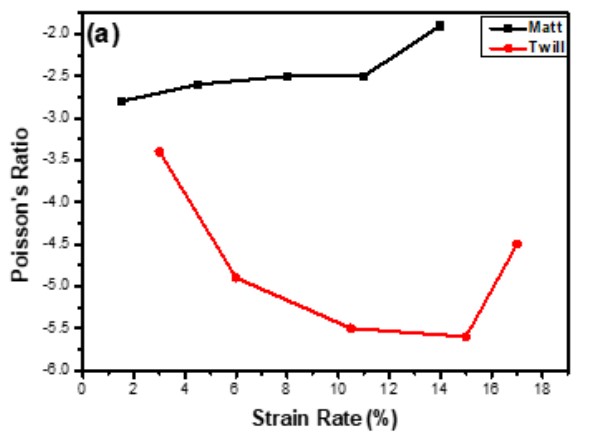 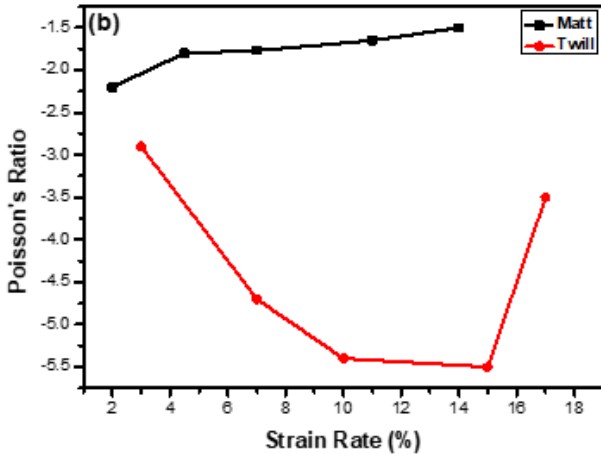

**Figure 11.** Graph between Poisson' Ratio and Strain percent for twill and matt fabrics (**a**) Warp wise, and (**b**) Weft wise.

Figure 11b shows the changes along the weft direction for both types of fabrics. The same increasing NPR trend with increasing strain rate is obtained for twill fabric while decreasing value of NPR for matt fabric is exhibited. The NPR values in out-of-plane auxeticity measurements are much higher, e.g., −5.6 for twill and −2.8 for matt along weft direction compared to the in-plane auxeticity tests due to the foldability and accumulation of flexible fabric in the form of wrinkles thus increasing the thickness of the fabric structure. The corresponding error bars graph is shown in Figure 12.

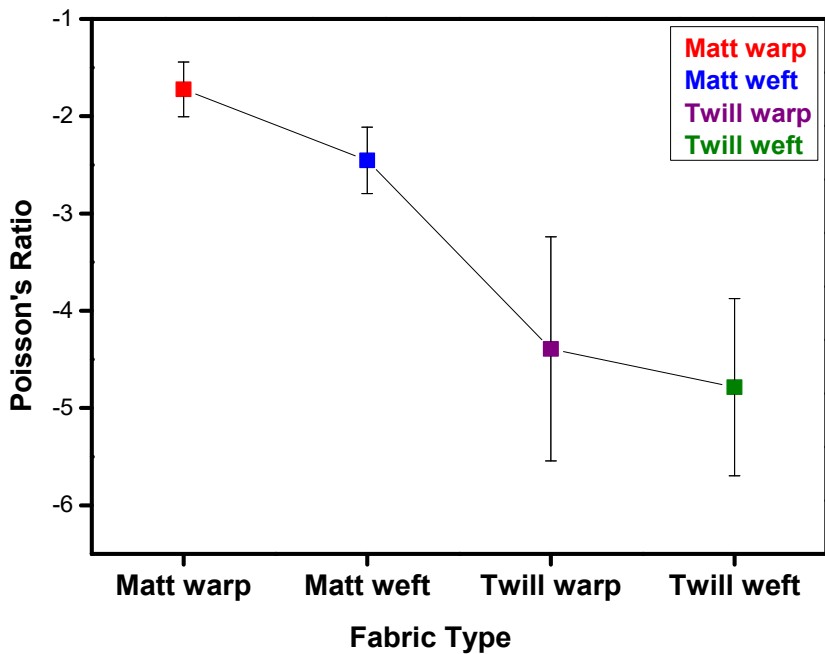

**Figure 12.** Out-of-plane Auxetic effect for all fabric samples having Poisson's ratio with error bars.

*3.5. Puncture Resistance Test*

The puncture strength of twill and matt woven fabrics was observed as 1.6 KN and 1.4 KN, respectively, as shown in Figure 13. The matt fabric showed 12.5% lower puncture strength and 16% less extension to puncture. Initially, when the stylus is not in touch with the fabric, both samples observe zero puncture strength. The curves start rising once a threshold value (here it is 10 mm) is achieved and the stylus touches the fabric surface. The compactness and higher interlacement of matt fabric restrict the movement of yarns at higher loads and therefore the breakage of yarn was observed on comparatively lower loads. Initially, when the stylus is not in touch with fabric, puncture strength curve is zero for both samples. However, when the stylus touches the fabric, the curves start rising. The uniformly compact structure and greater number of intersections in specific area and jammed structure of matt woven fabric make it a suitable candidate for puncture resistance while in twill it is easy to pass through lesser jammed/high float part. Inter-fibre friction is low due to lesser interlacements in lower interlaced high float twill fabric; therefore, stylus can push the yarns aside easily. Puncture tolerance of twill at higher strain is due to its higher extension. A major reason for lower strain tolerance in matt fabric is the lower extension in straight warp yarns, which cannot reorient themselves along with puncturing media. Although matt fabrics have good puncture resistance capabilities at lower loads, these are not suitable for ballistic protective applications where higher elongation are required. The ultimate puncture force is different for both the geometries, but the absorption energy is same because during extension of warp, weft yarns also contribute to the puncture energy absorption. Based on this test, twill design is not suitable for applications such as ballistics, where higher elongation is required. However, it is quite suitable for elastomeric composites-based applications, where tolerance to a higher extension before puncture is of critical importance. Matt fabric is preferred for such specific ballistic applications.

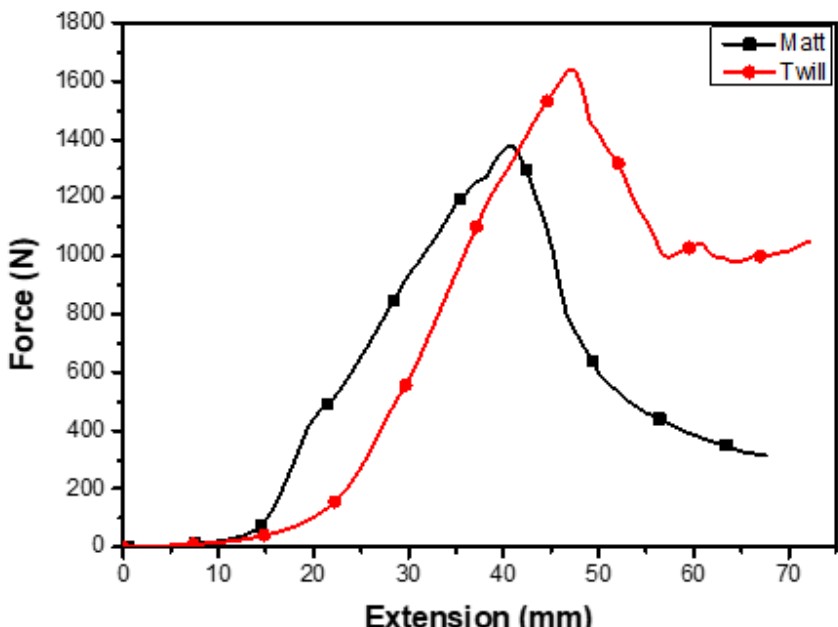

**Figure 13.** Puncture resistance test for twill and matt woven fabrics in warp direction.

*3.6. Energy Absorption/Impact Test*

The difference of the kinetic energy of initially fired pallet, and the final energy absorbed by the fabric produced the energy absorption by the fabric. The shot of air gun has an initial velocity of 140 m/s. The initial kinetic energy at the start of the test was measured as 98 Nm. The multiple fire tests were performed for both the fabric samples at different points on the fabrics as shown in Figure 14. Distortion is observed in the twill fabric while matt fabric shows little penetration. The matt fabric retards the pallet and higher penetrating velocity is observed due to fabric geometry where the yarns are closer to each other and the cross over points are nearer thus offer more resistance to the shot of air gun to pass easily through the fabric. Initially the fabric Auxetic actuation dissipates energy and subsequently the fabric has more chances to stop the impactor, thus enhancing its impact energy absorption. The excessive intersection points of matt weave pattern, dissipate less energy as compared to the twill weave structure having less intersection. On the other hand, larger float in the twill makes it easy for the shot to pass through the fabric with less energy Table 5 below shows the data for the energy measurements. It is obvious that more energy amounting to 84 Nm is absorbed by the matt fabric thus dissipating only 14 Nm compared to the twill fabric where 64.70 Nm of energy is absorbed, and 33.3 Nm was dissipated. These results are also in accordance with the already published literature [36,37], which reveals that the plain and matt design fabrics have higher energy absorption capacities compared to twill and satin designs. The tested fabric is not highly ruptured thus the fabric could withstand high energy.

Afs have several applications in the field of medical, sports, filters, protective textiles, and defence such as blast curtains. Medical textiles include drug delivery bandages, health monitoring sensors and compression garments. The extra ordinary higher energy absorption capabilities make Afs preferential candidate for sports such as racing, riding, and skating to ensure higher protection levels. Similarly, blast proof curtains achieve enhanced performance when made from Afs. High impact absorption, synclastic and deformability characteristics make the Afs suitable for other hi-tech defence applications such as nose cone in the aero planes. The ability of Afs to better fit bodies is another important aspect in their use for fashion garments and complex-shaped body parts.

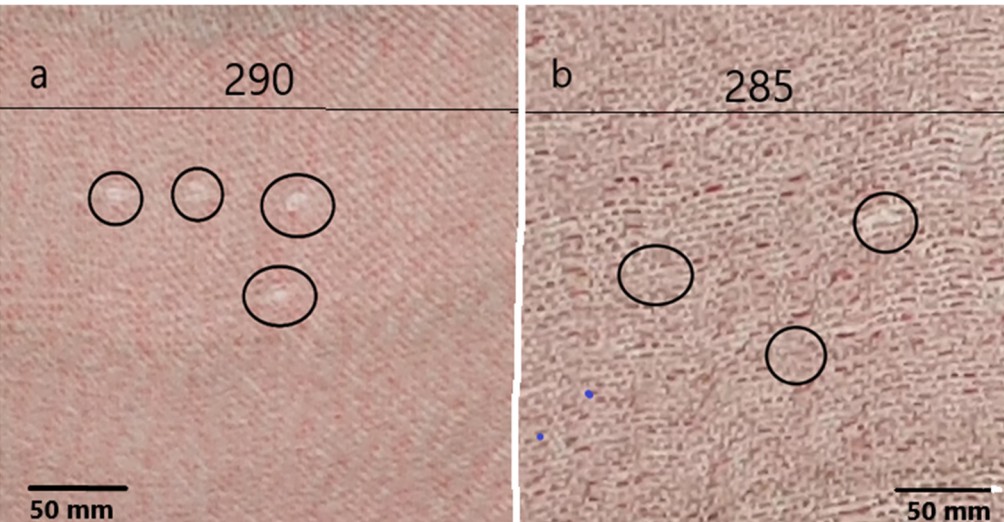

**Figure 14.** Fabric penetration during energy absorption test. (**a**) Twill fabric, and (**b**) matt fabric.

**Table 5.** Energy absorbed by matt and twill fabrics.

| Initial Energy Absorbed (Nm) K.E.=$\frac{1}{2}$mv$^2$ | Matt Fabric | | Twill Fabric | |
|---|---|---|---|---|
| | Velocity after Passing through Fabric (m/s) | Energy Absorbed (Nm) | Velocity after Passing through Fabric (m/s) | Energy Absorbed (Nm) |
| | 129.73 | 84.15 | 111 | 61.60 |
| 98 | 130.07 | 84.60 | 114.98 | 66.10 |
| | 129.27 | 83.55 | 115.23 | 66.40 |
| Average Values | 130 | 84 | 114 | 65 |
| Energy Discarded | | 14 | | 33 |

## 4. Conclusions

Various properties of the developed woven fabrics based on geometrical structure from the auxetic yarn with maximum NPR effect, were studied. The experimental results of out-of-plane and in plane auxetic behaviour in warp and weft directions of 3/1 twill and 2/2 matt woven fabric revealed higher NPR values of −5.6 and −2.17 for twill fabrics compared to lower values of −1 and −0.69 in matt fabric, respectively, due to increased fabric thickness because of accumulation of wrinkles in twill structure having long float lengths that provide more freedom of the auxetic yarns and offer less restriction in movement thus having more NPR effect.

The lower initial loads and more elongation at break make the matt fabric leading to superior protective nature of woven auxetic fabrics with higher impact energy absorption capabilities of 84 Nm as compared to twill auxetic fabric having absorption energy of 65 Nm with more structural stability for matt design than twill pattern fabrics.

The puncture resistance tests exhibited satisfactory results of 12.5% lower required puncture strength and 16% less extension to puncture for matt fabric due to high packing density but at higher loads they cannot sustain higher elongation and were not considered to be good candidates as protective fabrics.

The comparison of both design patterns reveals that the tests, 2D woven matt design structures exhibited superior properties such as in-plane NPR effect, energy absorption and puncture resistance while in other cases the twill fabric showed better properties such as out-of-plane auxeticity.

**Author Contributions:** Methodology, N.A.S.; Software, A.A.S.; Formal analysis, N.A.S. and M.I.; Data curation, A.A.S. and Y.N.; Writing—original draft, A.A.S.; Visualization, Y.N.; Supervision, M.S. All authors have read and agreed to the published version of the manuscript.

**Funding:** This research received no external funding from any source.

**Data Availability Statement:** This study includes all self generated data from experimentation. No data was reported from any external source.

**Acknowledgments:** We thank the Higher Education Commission of Pakistan for an International Research Support Initiative Program (IRSIP) scholarship to support A.A.S. We are also thankful to National Textile University Faisalabad, Pakistan for providing full access to their laboratory to perform the various experiments.

**Conflicts of Interest:** The authors declare no conflict of interest. The sponsors had no role in the design of the study; in the collection, analyses, or interpretation of data; in the writing of the manuscript, and in the decision to publish the results.

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
