# Peer review of "Effect of Geometric Arrangement on Mechanical Properties of 2D Woven Auxetic Fabrics"

_textiles, doi:10.3390/textiles2040035_

Round 1

Reviewer 1 Report

This manuscript discusses about the preparation of specialised fabric using polymers. The testing tools images, positioning of the samples during tests, various textile characterization were performed with the prepared fabric sufficiently. The results are useful for the modern textile industries. The results are explained with details in the discussion section. The explanations are readable. Thus it can be accepted in textiles journal.

1.      Line 131: Briefly give the detail of preparation method.

2.      Fig. 4: mention S1 to S7 inside the graph.

3.      Provide the photographic images of the sample textiles (with possible enlargement) prepared by warp, twill and other patterns.; Also mention the warp and weft directions.

4.      Fig. 8: Whether the samples belongs to the mentioned samples in table2 or different samples as given in table 3?.

5.      Mention the fabric production material (whether the PET+Polythene or PU+Polyester) combo wherever possible.

Reviewer 2 Report

Comments

In this paper, the authors studied tensile property, auxeticity, puncture resistance, and impact energy absorption capabilities of the developed woven fabrics based on geometrical structure from the auxetic yarn with maximum NPR effect. The purpose of the experiment is clear and the content is substantial. However, there are still some issues to be addressed. The specific comments can be found as following:

1、              The values in the tables in this paper should keep the accuracy of the same parameter consistent. For example, Poisson's value in Table 1 can be uniformly accurate to the thousandth; The Axial ΔL in Table 3 can be uniformly accurate to the hundredth and so on.

2、              The lines in Figure 4, Figure 5, Figure 6, Figure 9, Figure 10 and Figure 11 should be marked on the right side of the page.

3、              In Figure 8 and Figure 12, appropriate gaps should be left between the pictures to facilitate the distinction.

4、              There are still some typos and grammar issues in the manuscript. Authors should carefully recheck the whole manuscript.

5、              What is negative Poisson's ratio (NPR)? Could the authors introduce it in detail?

6、              Does geometric arrangement affect the hygroscopic property of 2D woven auxetic fabric?

Reviewer 3 Report

This work provides experimental results about the effect of the geometry arrangement on the mechanical properties of 2D woven auxetic fabrics. The work complements previous research by the authors in Ref. 29. and is relevant to the journal Textiles. However, the following comments should be addressed before the manuscript can be recommended for publication:

1. The abstract is too long. Please reduce it to 150-200 words maximum. Furthermore, a lot of the information in the abstract is not relevant. i.e., not related to the results and content of the paper.

2. Figures 2 and 3 do not show anything relevant or essential. It is just images of machines. Either modify to show, for example, the specimens being tested in the machine (or something relevant) or remove them from the manuscript.

3. Figure 7 does not show much. Zoom in to show the sample and the differences between initial and stretched positions.

Reviewer 4 Report

The manuscript on the topic Effect of Geometric Arrangement on Mechanical Properties of 2D Woven Auxetic Fabrics is an interest to the reader and within the scope of the Textiles MDPI. A couple of good findings can be observed such as the NPR of the auxetic yarns was dependent upon the selection of the core and wrap materials and the braiding angle. The NPR value was higher for a core with higher elasticity (e.g., PU elastomer compared with polyester); and a lower wrap angle with lower braiding speed exhibited higher NPR. However, the still manuscript needs revision in order to improve the overall quality of the script.

1.      The abstract section should be rewritten again. The abstract should state briefly the purpose of the research, the principal results, and major conclusions. Numerical values for the most important findings should be reported.

2.      On page 2 author have mentions “Auxetic honeycomb (AHC) exhibit 43 best auxetic nature amongst all auxetic geometries” Kindly justify it.

3.      Author mentions “Auxetic textiles have attracted great attention in the recent era. Researchers have tried to induce the auxetic behaviour from fibre to fabric level with auxetic geometries in the textile structures”. Kindly briefly provide some of the related research from the literature.

4.      Kindly provide the motivation behind this work.

5.      Authors should provide a statement describing one or more key hypotheses that the work described in the manuscript was intended to confirm or refute. The inclusion of a hypothesis statement makes it simple to contrast the hypothesis with the most relevant previous literature and point out what the authors feel is distinct about the current hypothesis (novelty).

6.      Authors should write text in good English (American or British usage is accepted, but not a mixture of these). English language of the manuscript required editing to eliminate possible grammatical or spelling errors and to conform to correct scientific English.

7.      On page 5, section 2.2.1. Characterization of the woven fabric, provide the made and operating condition of all the instruments used in the current work.

8.      On page 6, Figure 4. Maximum NPR vs braid angle for samples S1 to S7. Kindly add the errors bar in the graph.

9.      Figure 5. Comparison of tensile properties of auxetic woven fabric (a) Matt weft and warp, (b) Twill  weft and warp. This fig. 5 discussion is very confusing. Kindly authors should explain the discussion of the fig.5a,,b which can easily be understand to the reader.

10.  On page 15, Figure 10. Graph between Poisson’ Ratio and Strain percent for Twill and Matt fabrics (a) Warp 514 wise, and (b) Weft wise . Provide with error bar.

11.  Conclusion section needs to be improved and shortened. The Conclusion should not be a summary but should illustrate the advances and claims of innovative aspects of the research work done.

Round 2

Reviewer 2 Report

Accept in present form.

Reviewer 4 Report

The authors have satisfactorily revised the manuscript.